# Using resistor network models to predict the transport properties of solid-state battery composites

Lukas Ketter ®[1,2], Niklas Greb[1], Tim Bernges[1] & Wolfgang G. Zeier ®[1,2,3] ✉

Solid-state batteries use composites of solid ion conductors and active materials as electrode materials. The effective transport of charge carriers and heat thereby strongly determines the overall solid-state battery performance and safety. However, the phase space for optimization of the composition of solid electrolyte, active material, additive is too large to cover experimentally. In this work, a resistor network model is presented that successfully describes the transport phenomena in solid-state battery composites, when benchmarked against experimental data of the electronic, ionic, and thermal conductivity of $LiNi_{0.83}Co_{0.11}Mn_{0.06}O_2$-$Li_6PS_5Cl$ positive electrode composites. To highlight the broadness of the approach, literature data are examined using the proposed model. As the model is easily accessible and expandable, without the need for high computing power, it offers valuable guidance for experimentalists helping to streamline the tedious process of performing a multitude of experiments to understand and optimize the effective transport of composite electrodes.

Lithium-ion batteries are an inevitable part of energy storage in our modern world. However, conventional lithium-ion batteries are expected to run into performance limits[1]. To push battery performance beyond these limitations, it is critical to investigate alternative battery technologies. One particularly promising technology is solid-state batteries (SSB). By replacing separators and liquid electrolytes of lithium-ion batteries with solid-state ion conductors, the usage of Li-metal negative electrodes and thus higher energy densities can potentially be enabled[2–4]. Further advantages are increased mechanical stability, the absence of leakage and a potentially improved thermal safety[5].

Solid-state batteries utilize composite electrodes, consisting of the electrochemically active material, the solid electrolyte (SE) and, if necessary, additives such as binders or conductive carbons, e.g., vapor grown carbon fibers (VGCF). In these composite systems, a sufficiently high ionic ($\sigma_{ion}$) and electronic ($\sigma_e$) conductivity are necessary, given that both charge carriers need to access the active material during the charging and discharging reactions. Since a large mismatch in effective

ionic and electronic conductivity leads to inhomogeneous reaction rates throughout the whole electrode thickness, balancing of the two transport quantities is of paramount importance to enable high electrode utilizations and avoid reaction fronts during battery operation[6,7].

On the one hand, charge carrier transport can be decisively influenced by varying the volumetric ratios of the electrode components given that the electrochemically active material and carbon additives are solely responsible for the electronic conduction, while the solid electrolyte exclusively conducts ions[8,9]. With that, altering the SE content in composite anodes, such as $Li_4Ti_5O_{12}$-$Li_7P_2S_8I$-C[10] or Si-$Li_6PS_5Cl$ (LPSCl)-C[11], can facilitate changes in the effective ionic conductivity over orders of magnitude. Similarly strong correlations between effective ionic and electronic conductivity and the volumetric ratios of the composite components were found in cathode systems such as NCM622-LPSCl[12], $LiMn_2O_4$-$Li_3InCl_6$[13] and S-LPSCl-C[14]. On the other hand, the microstructure of the composite electrode can influence its effective conductivities. Froboese et al.[15] demonstrated the influence of differently sized inclusions on the effective ionic

[1]Institute of Inorganic and Analytical Chemistry, University of Münster, 48149 Münster, Germany. [2]International Graduate School of Battery Chemistry, Characterization, Analysis, Recycling and Application (BACCARA), University of Münster, 48149 Münster, Germany. [3]Forschungszentrum Jülich GmbH, Institute of Energy and Climate Research Helmholtz-Institute Münster (IMD-4), 52425 Jülich, Germany. ✉e-mail: wzeier@uni-muenster.de

conductivity of composites. Studies investigating the effect of changing active material particle sizes in NCM622-LPSCI[16] and Si-LPSCl-C[17] electrodes on the battery performance highlight the importance of carefully designing microstructures to improve electrode utilizations. In a similar manner, optimizing the particle size of the SE in $LiNi_{0.83}Co_{0.11}Mn_{0.06}$ (NCM83)-LPSCl[18] composites leads to better electrode utilizations due to a more homogeneous ion current distribution during battery operation. This emphasizes that both the active material and electrolyte particle size need to be considered in electrode design.

While the influence of volume fractions and microstructure on charge carrier transport is frequently investigated, the influence on the thermal transport properties is scarcely studied[19–21]. Nonetheless, due to internal resistances ($R$) within the solid-state battery system, Joule heating occurs during battery operation. This is of particular importance during fast charging, as the power of heating ($P$) is related to the square of the applied current ($I$) via $P = R \cdot I^2$ [22]. When the generation of thermal energy is faster than its dissipation, heat accumulates in the battery system leading to an increase of the battery temperature and the development of temperature gradients[23]. This heating and temperature evolution within a cell can significantly impact battery performance, accelerate degradation kinetics and lead to at-times violent device failure[24,25]. Particularly considering battery safety, thermal transport is known to play a critical role in conventional batteries[22,26]. Even though solid-state batteries are often promoted as being safer than their conventional Li-ion counterparts, potential safety concerns should not be overlooked and carefully studied. Using thermodynamic models, Bates et al.[27] showed, that under short-circuit conditions larger temperature increases are possible in SSB compared to batteries with liquid electrolytes. Furthermore, severe reactions and fire generation even under inert atmospheres were observed by Kim et al.[25,28] when they exposed charged cathode composites to elevated temperatures above 150 °C. This is further corroborated by exothermic reactions that were observed between SSB-components in a differential scanning calorimetry study by Johnson et al.[29] Especially regarding the thermal management design, thermal transport considerations for SSB pose an essential addition to the often-conducted ionic and electronic conductivity studies. Thermal transport investigations on various SE have shown that their thermal conductivities are low and show a "glass-like" temperature-dependence[21]. Density dependent thermal transport investigations on sulfidic Li+[30] and Na+[31] conducting SE revealed further lowering of the thermal conductivity with porosity, leading to thermal

conductivities in the range of liquid electrolytes. The low thermal conductivity of SE materials indicate that slow heat dissipation and potential thermal runaway warrant careful consideration.

Generally, evaluating transport in composites as a function of composition relies on a variety of measurements, making them a time-consuming endeavor. These measurements are tedious considering that processing, additives, binders, and any microstructural changes will influence the effective transport through the composite and that transport needs to be re-evaluated every time. Aside from experimental approaches, microstructure-based simulations can offer valuable guidance when developing microstructural design concepts. Considering the meso- and even atomistic scale, Heo et al.[32] investigated the impact of microstructure on ionic conductivity in $Li_7La_3Zr_2O_{12}$ computationally, while Haruyama et al.[33] calculated optimized structures and properties of interfaces between oxidic cathode active material (CAM) and $\beta$-$Li_3PS_4$ using density functional theory. Microstructural modelling of entire composite cathodes can further reveal percolation behavior of ion- and electron-conducting clusters[8], while effective electronic and ionic conductivities can be calculated using flux-based simulations on composite microstructure representations[9,34]. Finsterbusch et al.[35] constructed a virtual twin based on scanning electron microscopy micrographs and subsequently simulated discharge curves. Extending this approach, Chen et al.[36] utilized an electrochemical-thermal coupled model to simulate heat evolution in addition to discharge characteristics of composite cathodes. Nevertheless, these approaches are often computationally expensive, and a faster predictive approach is desirable to screen the transport of new systems.

In this work, we propose a simple resistor network model describing transport in solid-state battery electrode composites based on simple voxel structures that is then benchmarked against experimental transport in NCM83-LPSCl cathode composites. A schematic two-dimensional representation of a typical solid-state composite cathode microstructure in comparison to a voxel microstructure analogue is shown Fig. 1. Despite this simplification, essential concepts like tortuosity and domain size are reflected by the model. Concluding the experimental characterization and microstructure-based transport simulations of the ionic, electronic, and thermal transport properties of the NCM83-LPSCl system, the proposed microstructural transport model is tested against literature data to show its capability to predict transport of other composite systems. The results show that the model can easily be modified and extended to other systems and research questions. Using the resistor network model provides experimentalists a new tool to fast find the range of optimized solid-state battery compositions – without the need for much computational resources.

## Results

### Resistor network modeling of solid-state battery composites

When a material is subjected to a temperature gradient ($\nabla T$), heat flow occurs. According to Fouriers law (Eq. 1) the heat flux density ($q$) is linked to the temperature gradient by the thermal conductivity ($\kappa$) of the material[37]:

$$q = -\kappa \nabla T \tag{1}$$

An equivalent relation for the movement of charge carriers (ions and electrons) as response to an electric field is given by Ohm's law (Eq. 2), following[38]:

$$J = -\sigma \nabla \phi \tag{2}$$

where $J$, $\sigma$ and $\nabla \phi$ are the electrical flux density, the electrical conductivity and the potential gradient, respectively. In the resistor network model proposed in this work we make use of these fundamental equations by applying a virtual gradient to the voxel representations of

**Composite schematic**

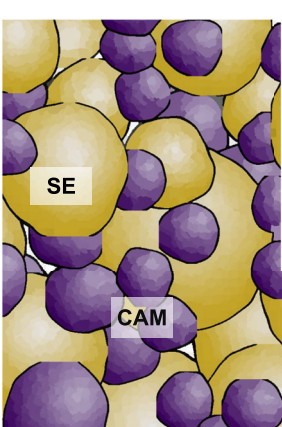

**Voxel representation**

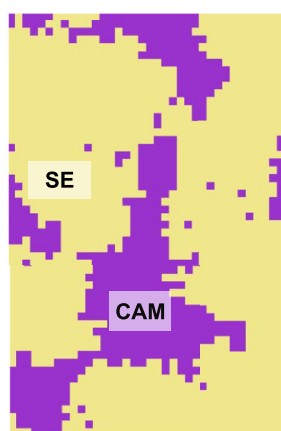

**Fig. 1 | Schematic and voxel representation of a composite electrode.** Comparison between a schematic composite electrode and a top view on a voxel representation of a composite consisting of cathode active material (CAM) and solid electrolyte (SE).

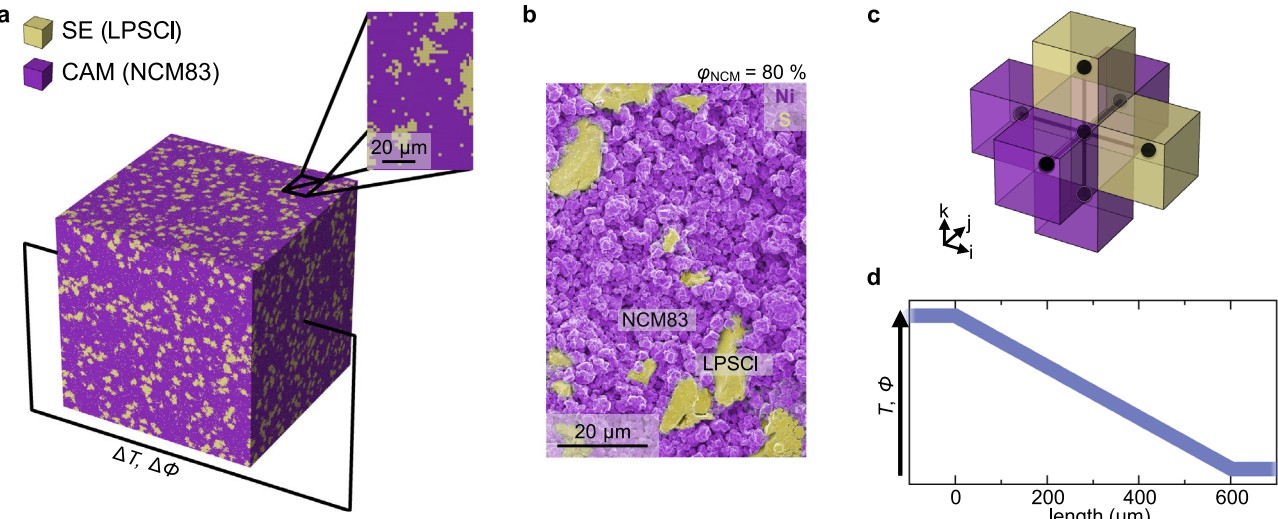

**Fig. 2 | Principle of the resistor network modelling approach. a** Voxel representation of a composite with a NCM83 volume fraction of $\varphi_{NCM} = 80\%$ and a LPSCl volume fraction of $\varphi_{LPSCl} = 20\%$ generated with 300 x 300 x 300 voxels. **b** Representative false-colored SEM micrograph of a composite with $\varphi_{NCM} = 80\%$ and $\varphi_{LPSCl} = 20\%$. Voxels and areas colored yellow correspond to LPSCl, whereas purple areas correspond to NCM83. **c** A representative voxel with its adjacent voxels, representing the resistor network structure and its nodes. **d** Schematic of the resulting average temperature and potential distribution along the direction of heat- and current flow in the steady state.

our composite electrodes, compute the steady states and derive effective transport properties[39–42].

The workflow can be described as follows (detailed description in Section S1 of the Supplementary Information): First, a microstructure consisting of 300 x 300 x 300 cubic voxels is constructed. Each voxel represents a 2 μm x 2 μm x 2 μm domain and is assigned to either LPSCl or NCM83, leading to a total microstructure edge length of 600 μm (Fig. 2a). The number of voxels assigned to each phase is determined by the volume ratio that is investigated. A representative false-colored scanning electron microscopy micrograph of a composite is shown in Fig. 2b. In this composition, LPSCl domains ($d \approx 20$ μm) are embedded in a matrix of smaller NCM83 particles ($d \approx 2$ μm). To account for the larger solid electrolyte particles in the microstructure assembly, clustering is applied for voxels representing LPSCl (Figure S1).

Next, a resistor network is generated based on the voxel structure (Fig. 2c) considering that each voxel has either the transport properties of the SE or the CAM. With that, the conductivities of the pure phases are input parameters in the model. Each voxel center corresponds to a node in the resistor network with temperature $T_{i,j,k}$ or potential $\phi_{i,j,k}$, respectively, whereas the indices $i$, $j$ and $k$ describe the node positions in the three-dimensional structure. Each node (e.g., at position $i$, $j$ and $k$) is linked to adjacent nodes (e.g., $i+1$, $j$ and $k$) by resistors, and the conductivity between the nodes is determined by the properties of the connected voxels. Subsequently, a virtual constant temperature or potential gradient is applied to two opposing surfaces, while zero-flux boundaries are employed at the other surfaces of the resistor network (Figure S2a) and the node temperatures or potentials are iteratively adjusted until the steady state condition is sufficiently met. From the steady state temperature or potential distribution (Fig. 2d), the effective conductivities are computed by calculating the average flux density going through the resistor network and by solving Eq. 1 or Eq. 2 for the respective conductivity in one dimension.

### Exploring the electrical transport properties of solid-state battery composites

The effective ionic and electronic conductivities were measured as a function of the volume fractions of NCM83 ($\varphi_{NCM}$) and LPSCl ($\varphi_{LPSCl}$) using electrochemical impedance spectroscopy (EIS) and direct current (DC) polarization (details in Section S4 of the Supplementary Information). Ion-blocking contacts are employed to analyze electronic conductivities, while electron-blocking conditions are chosen to assess the ionic conductivities (Fig. 3a). A transmission line model (TLM), previously established[12] for the related NCM622-LPSCl composite system is employed to analyze the impedance results (Fig. 3a). The TLM serves as an equivalent circuit describing the electrical transport of composite cathodes by interconnected ion- and electron-conducting paths. Although not providing a complete and accurate description of the complex underlying charge transport in this composite class, it well reflects many aspects of the contemporary understanding about charge transport in electrode composites and, more importantly, can be used to correctly determine the total effective conductivities from the impedance spectra[12]. Exemplary experimental results and TLM fits of a NCM83-LPSCl composite cathode with $\varphi_{NCM} = 40\%$ are shown in Fig. 3b, respectively. To further verify the ability of the TLM to determine the desired transport properties, in section S4.2 an alternative equivalent circuit often used for mixed ionic-electronic conductors[43] is used to evaluate the impedance data of the composite with $\varphi_{NCM} = 40\%$.

To complement the impedance results, DC polarization experiments were conducted in which the current response to different applied voltages is evaluated. Depending on whether ion- or electron-blocking contacts are employed, the resulting current can be assigned exclusively to the transport of electrons or ions, respectively. As current and the applied potential difference are linked via Ohm's law (Eq. 2), the effective electronic or ionic conductivity can be determined accordingly. An exemplary dataset for $\varphi_{NCM} = 40\%$ is shown in Fig. 3c.

After the typical experimental characterization, the effective ionic and electronic conductivities are separately calculated using the resistor network model. In the model, the experimentally assessed values for $\sigma_{ion,LPSCl}$ and $\sigma_{e,NCM}$ were taken as input to describe the transport of voxels representing LPSCl and NCM83. Additionally, LPSCl and NCM83 were considered exclusively ion- and electron-conducting, respectively, thereby assuming $\sigma_{e,LPSCl} = \sigma_{ion,NCM} \approx 0$ mS cm$^{-1}$. This simplification does not significantly influence the obtained effective conductivities given that $\sigma_{ion,NCM}$[44] and $\sigma_{e,LPSCl}$[45] are orders of magnitude below $\sigma_{ion,LPSCl}$ and $\sigma_{e,NCM}$ measured in this work.

The measured and simulated effective electronic and ionic conductivities are in good agreement as shown in Fig. 3d. In line with

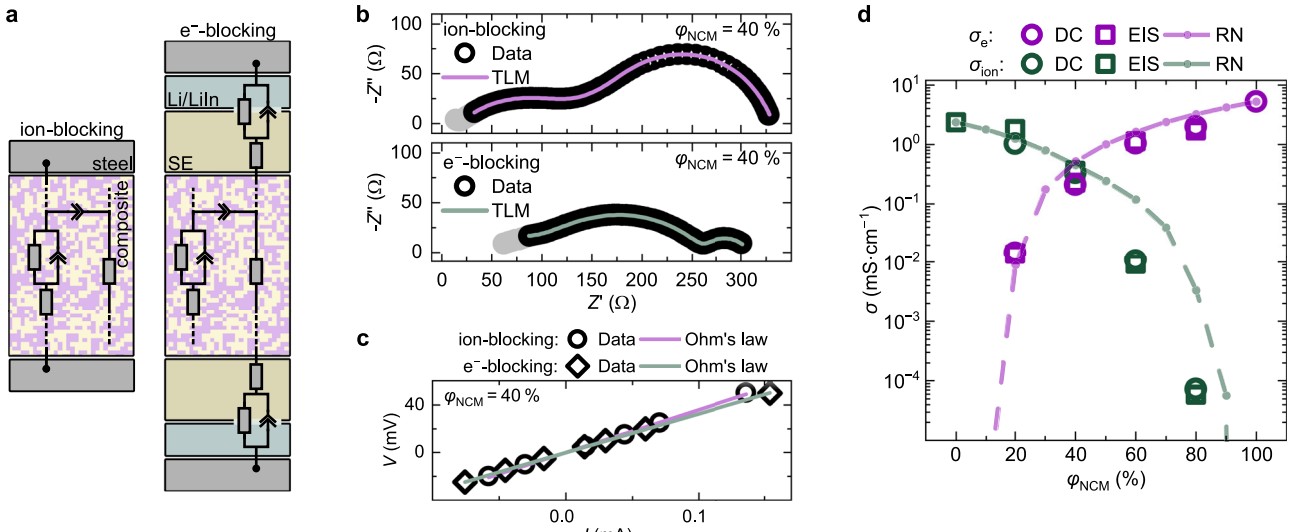

**Fig. 3 | Electrical transport in composite electrodes. a** Schematic setups used to ensure ion-blocking and electron-blocking measurement conditions respectively together with transmission line models (TLM) used to evaluate impedance data. **b** Electrochemical impedance spectroscopy data (circles) measured under ion- and electron-blocking conditions are exemplary shown for a NCM83-LPSCl composite with a NCM volume fraction of 40 %. Data colored in grey are not considered during the fit (line). **c** Direct current (DC) polarization data measured under ion-blocking (circles) and electron-blocking (diamonds) conditions respectively, as well as the

corresponding fits (line) are exemplary shown for a NCM83-LPSCl composite with a NCM volume fraction of 40 %. **d** Resulting effective ionic- and electronic conductivities as a function of NCM volume fraction. Effective electronic conductivities measured via DC polarization (circles) and EIS (squares) are shown in purple, whereas effective ionic conductivities are depicted in green. Each data point corresponds to a single measurement. Effective conductivities from simulations with the resistor network model are shown as dots connected with straight dashed lines as guide to the eye.

previous results[12], the effective electronic and ionic conductivities change over orders of magnitude when the volume ratio of NCM83 to LPSCl in the composites is varied. While the effective electronic conductivities are in the range of $10^1$ mS cm$^{-1}$ to $10^{-2}$ mS cm$^{-1}$, the effective ionic conductivities are in the range of $10^1$ mS cm$^{-1}$ to $10^{-5}$ mS cm$^{-1}$ for the investigated materials. The effective ionic and electronic conductivity are closest to each other for a composition of $\varphi_{NCM} = 40$ % and a dramatic drop in the effective ionic and electronic conductivity is observed when approaching low volume fractions of LPSCl and NCM83, respectively. This behavior is also reflected by the resistor networks and is expected given the loss of percolation pathways of the respective conducting phases. Since the percolation threshold strongly depends on the microstructure, larger deviations between measured and simulated conductivities can be expected close to the threshold as only simplified, virtual microstructures are assumed in the resistor network simulations[42,46]. Differences in the measured and simulated effective ionic conductivities for compositions with $\varphi_{NCM} > 40$ % may be attributed to this effect.

Additionally, the influence of VGCF additives on effective electronic and ionic conductivity is investigated for a composite with $\varphi_{NCM} = 40$ % and discussed in Section S2 of the Supplementary Information. While the electronic conductivity increases over two orders of magnitude upon introduction of <2 wt.% VGCF, only minor influences on the ionic conductivity are observed.

## Exploring the thermal transport properties of solid-state battery composites

Besides the electronic and ionic transport, the heat transport of composites will be decisive for the safety of a solid-state battery, and the proposed resistor network model needs to account for heat transport, too. To investigate and compare the thermal properties of NCM83, LPSCl and their composites, thermal diffusivities (D) were determined experimentally. The thermal diffusivity is a measure of how quickly temperature spreads through a material. Exceptionally low thermal diffusivities ranging from 0.18 mm$^2$ s$^{-1}$ to 0.43 mm$^2$ s$^{-1}$ were measured for all compositions (Figure S15).

In this work, isobaric heat capacities ($C_P$) are approximated by calculated isochoric heat capacities (details in the Supplementary Information Section S6). Within the investigated temperature range, all heat capacities show a steady increase with temperature (Fig. 4a) following the excitation of more vibrational modes with increasing temperatures. While LPSCl shows heat capacities ranging from 0.72 J g$^{-1}$ K$^{-1}$ to 1.10 J g$^{-1}$ K$^{-1}$, NCM83 shows lower heat capacities from 0.39 J g$^{-1}$ K$^{-1}$ to 0.85 J g$^{-1}$ K$^{-1}$. By considering the measured thermal diffusivities, calculated heat capacities and geometrical densities (Figure S18), thermal conductivities ($\kappa$) are calculated via $\kappa = \rho \cdot D \cdot C_P$. The resulting average thermal conductivities of triplicate measurements are shown in the Fig. 4b, whereby the uncertainties represent the standard deviations. The thermal conductivities are remarkably low and remain approximately constant over the measured temperature range. The room-temperature thermal conductivities are shown as a function of $\varphi_{NCM}$ in Fig. 4c. LPSCl shows the lowest thermal conductivity in the series with 0.32 ± 0.02 W m$^{-1}$ K$^{-1}$. This thermal conductivity lies within the typical range for sulfidic solid-state electrolytes[21,30]. With a value of 0.71 ± 0.04 W m$^{-1}$ K$^{-1}$, NCM83 shows the highest thermal conductivity of the investigated series in agreement with the thermal conductivities of calendared NCM electrodes reported by Gandert et al.[47] An increase in effective thermal conductivity with increasing NCM83 volume fraction is observed.

To corroborate these experimental findings, the resistor network model is utilized to simulate the thermal conductivities of the composites based on the conductivities of LPSCl and NCM83. The simulated thermal conductivities overestimate the experimental data and going forward, interfacial thermal resistances are evaluated as potential origin for the mismatch.

At perfect interfaces between different materials, interfacial thermal resistances in the range of $10^{-9}$ m$^2$ K W$^{-1}$ to $10^{-7}$ m$^2$ K W$^{-1}$ are caused by a mismatch in phonon structure. For imperfect interfaces at a macroscopic scale, additional effects such as porosity and surface roughness further enhance this resistance, whereas typical values lie between $10^{-6}$ m$^2$ K W$^{-1}$ to $10^{-3}$ m$^2$ K W$^{-1}$[48]. Using this range of interfacial resistances as guidance, it is found empirically that better agreement

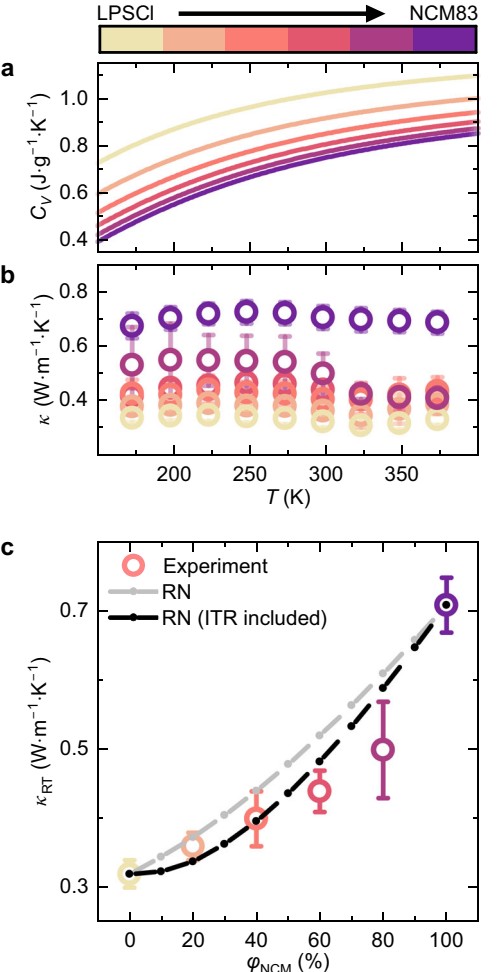

**Fig. 4 | Thermal transport in composite electrodes.** Data for LPSCl, NCM83 and NCM83-LPSCl composites are shown color-coded from yellow to purple with $\Delta\varphi_{NCM} = 20\,\%$. **a** Specific isochoric heat capacities in the temperature range from 173.15 K to 373.15 K. **b** Thermal conductivities in the temperature range of 173.15 K to 373.15 K. The data (circles) shown correspond to the mean of triplicate measurements with the error bars representing their standard deviations. **c** Experimentally assessed room temperature (25 °C) thermal conductivities (color-coded circles) as a function of NCM volume fraction in comparison to simulated thermal conductivities (dots connected with dashed lines as guide to the eye) with (black) and without (grey) considering interfacial thermal resistances (ITR). The measurement data correspond to the mean of triplicate measurements with the error bars representing their standard deviations.

between simulation and experiment can be achieved considering a interfacial thermal resistances of $2 \cdot 10^{-6}$ m² K W$^{-1}$ between the LPSCl and NCM83 domains in the resistor network model (Fig. 4c). This indicates the presence of interfacial thermal resistances between LPSCl and NCM83 providing a physically reasonable explanation for the slower increase of effective thermal conductivity with $\varphi_{NCM}$.

Additionally, the influence of VGCF additives on effective thermal transport is investigated for a composite with $\varphi_{NCM} = 40\,\%$ and discussed in Section S2 of the Supplementary Information. Despite the intrinsically high thermal conductivity of carbon fibers[49], only minor influences on the effective thermal conductivity on the composites are observed in this work.

With that, this work shows that the heat transfer in the NCM83-LPSCl composite system is slow with thermal conductivities below 1 W m$^{-1}$ K$^{-1}$, independent of the volumetric ratio and VGCF introduction, and that based on our resistor network analysis, interfacial

thermal resistances might further lower the thermal conductivity of the composites.

## Applying the resistor network model to other solid-state battery research questions

Resistor network models are widely used in various fields such as statistical physics[50], thermal barrier coatings[39] or in fuel cells[51], to name just a few examples. To highlight the application space and predictive power of the proposed resistor network model in the case of solid-state battery composites, three literature cases are examined. An additional case study, as well as Table S3, listing the input parameters used to run the simulations are shown in section S9 of the Supplementary Information. In a study by Hendriks et al.[13], the effective ionic and electronic conductivities of LiMn$_2$O$_4$-Li$_3$InCl$_6$ cathode composites were investigated (Fig. 5a). As in the NCM83-LPSCl system discussed above, significant changes in terms of charge carrier transport are observed upon varying the volumetric ratio.

Analogous to the investigated system in this work, i.e., NCM83-LPSCl composites, the CAM (LiMn$_2$O$_4$) was assumed to be exclusively electron-conducting and the SE (Li$_3$InCl$_6$) exclusively ion-conducting. The effective $\sigma_{e,LiMn_2O_4}$ and $\sigma_{ion,Li_3InCl_6}$ reported by Hendriks et al.[13] were used as experimental inputs. The simulated effective conductivities are in good agreement with the experimental literature data (Fig. 5a), showing that the model is capable to describe the transport through composites of various chemistries.

Experimental results by Böger et al.[30] are investigated as a second case study that discuss the influence of macroscopic density (porosity) on the thermal conductivities of LPSCl (Fig. 5b). Hereby, the porous LPSCl can be considered as a two-phase system consisting of argon filled pores and solid electrolyte. Assuming $\kappa_{LPSCl} = 0.66$ W m$^{-1}$ K$^{-1}$ for the electrolyte[30] and 0.017 W m$^{-1}$ K$^{-1}$ as thermal conductivity of argon filled pores[52] as inputs for the resistor network model, effective thermal conductivities of porous LPSCl were simulated and are shown in Fig. 5b. Although slightly overestimating the reported effective thermal conductivities, the simulated data are generally in good agreement with the experimental results.

As a third study, the experimental results by Froboese et al.[15] are evaluated, who investigated the influence of differently sized ion-blocking inclusions on the effective ionic conductivities of a PEO:LiTSFI:SiO$_2$ – solid electrolyte matrix (Fig. 5c)[15]. It was shown that for the same volume fraction of inclusions, smaller inclusion particles lead to a more pronounced decrease in the effective ionic conductivity. This effect shows more drastically for higher volume fractions of the inclusions[15]. To analyze this behavior using the resistor network model, the reported ionic conductivity of the PEO:LiTSFI:SiO$_2$ – solid electrolyte matrix was taken as a starting parameter. Ion-blocking clusters of different sizes (Fig. 5d) were constructed and used to describe the influence on the effective ionic conductivity. The simulated effective conductivities are in good agreement with the results by Froboese et al.[15]. In agreement with the literature results, the amplified influence of size effects with increasing volume fraction and the introduction of smaller ion-blocking clusters is captured by the resistor network model. While the general trend is reflected well, a weaker decrease in conductivity beyond inclusion fractions of approximately 30 vol.% is predicted by the resistor network model. This deviation is likely related to the simplifications of the model and the differences between the actual microstructure and the constructed voxel structure (particle shape).

These case studies highlight that the resistor network model can guide the understanding of transport and the effect of particle size in various composites of different active materials and different inorganic and polymeric solid electrolytes. Thereby, the model offers a tool to accelerate the optimization of composites, either when new chemistries are investigated, or when microstructures are designed, by providing predictive power for experimentalists. Even though the

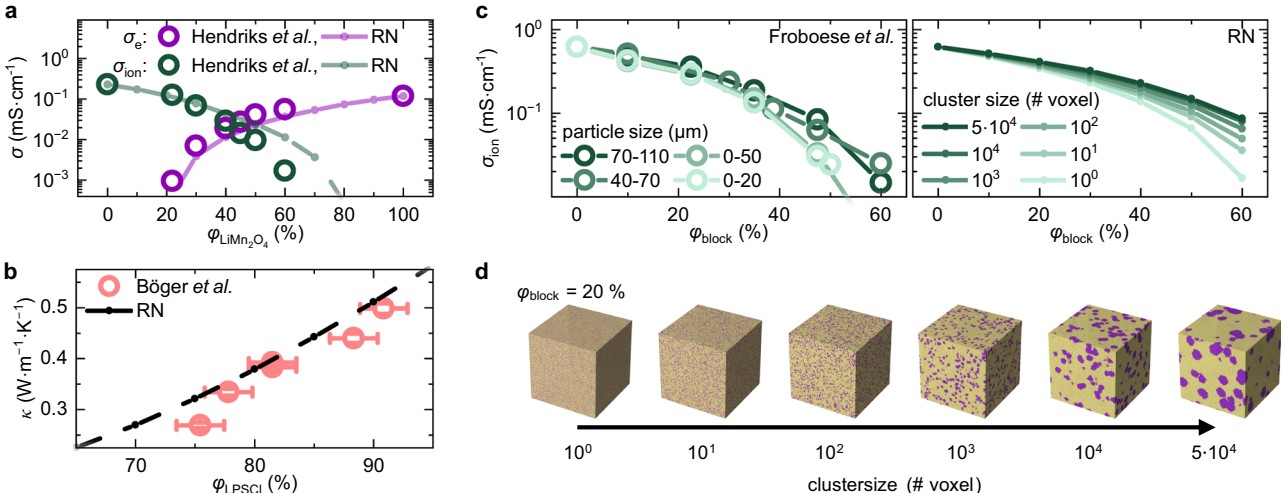

**Fig. 5 | Analyzing literature case studies using resistor network models. a** Ionic (green) and electronic (purple) conductivities reported by Hendriks et al.[13] (circles) in comparison to simulated conductivities of the resistor network model (dots with dashed guide to the eye) as a function of $LiMn_2O_4$ volume fraction. **b** Thermal conductivities reported by Böger et al.[30] (orange circles) in comparison to results of the resistor network model (dots with dashed guide to the eye) as a function of LPSCl volume fraction. **c** Influence of the particle size and volume fraction of an insulating second phase on the effective ionic conductivities as reported by Froboese et al.[15] in comparison to the results of the resistor network model using various cluster sizes and volume fractions ($\varphi_{block}$) of insulating phase show good agreement with the experimental results. **d** Exemplary voxel structures for inclusions of different sizes at an inclusion volume fraction of $\varphi_{block} = 20$ %.

large resistor networks computed in this work were calculated using high performance computing infrastructure, reproducible simulations can already be achieved with smaller networks (Figure S22), allowing experimentalists to compute a first guess regarding transport trends on a standard computer. For more advanced research questions such as composites without well-defined microstructures or additives that cannot be simply modelled by a voxel approach (Figure S21) however, the model runs into limitations. While the current model can predict macroscopic effective total transport properties based on composite microstructures, it does not yet provide detailed microscopic insights on charge transport mechanisms. With that, to capture complex electrode microstructures, occurring reaction pathways and chemo-mechanical influences in its complete detail, continuum models are still an important and highly valuable approach[9,34,35].

## Discussion

In this work, the ionic, electronic and thermal transport of NCM83-LPSCl composite cathodes has been studied experimentally. While the electronic and ionic conductivities change over orders of magnitude upon changing the volume ratios of NCM83 and LPSCl, the thermal conductivities of the composite cathodes show only minor changes with composition and are remarkably low throughout all investigated samples.

In this work, a resistor network model is presented to predict the ionic, electronic and thermal transport of NCM83-LPSCl composite cathodes and experimentally validated. The model approach helps to provide a deeper understanding of the experimental transport data. Voxel structures representing the composite microstructure were generated and transport properties derived from steady-state calculations. Despite its simplicity, the simulated effective conductivities agree well with the measured data in this work and literature case studies, showing that influences of percolation, interfacial resistances and domain sizes are reflected over different active materials and solid electrolytes.

Knowledge about effective transport is crucial for electrode design considerations, but requires time-consuming, and labor-intensive measurements. This model offers guidance for experimentalists and can be used to make basic predictions for composition-dependent transport. As the model can easily be extended to include

additional concepts such as particle size distributions and particle shapes, it can streamline transport investigations in electrode composites. The simplicity of the approach sets it apart from many transport models. Hence the objective is also different: While many transport models aim to reflect exact representation of real composites, which is of immense value to deeply understand transport in such systems, the model presented in this work aims to guide experimentalists in finding optimized solid-state battery composites fast when changing the materials and particle morphologies in solid-state battery research.

## Methods

Unless otherwise stated, all preparative work was carried out in a glovebox under argon atmosphere.

### Synthesis

LPSCl was synthesized via a solid-state synthesis route. Therefore LiCl (Sigma-Aldrich, 99%), $Li_2S$ (Thermo Scientific Chemicals, 99.9%) and $P_4S_{10}$ (Sigma-Aldrich, 99%) were weighed in stoichiometrically and hand ground for 15 min. The powder mixture was compacted using a hand press and placed in a carbon-coated quartz ampoule, which was then sealed under dynamic vacuum. After heating to 550 °C with a heating rate of 100 °C/h, the mixture was kept at this temperature for two weeks. After natural cooling to room temperature, the product was hand ground and used for further investigations.

### Soft milling

LPSCl, NCM83 (MSE supplies, dried overnight at 250 °C under dynamic vacuum) and mixtures of both materials were soft milled prior to transport measurements. For composites, LPSCl and NCM83 were combined in the respective volume ratios whereby 1.86 g cm$^{-3}$ and 4.78 g cm$^{-3}$ were assumed as densities[53,54]. Furthermore, composites with additional 1 wt.% to 5 wt.% of VGCF (Sigma-Aldrich, dried overnight at 250 °C under dynamic vacuum) were prepared. Together with spherical $ZrO_2$ milling media (3 mm diameter, 3.6 g), the combined materials (400 mg) were placed into $ZrO_2$ cups (15 mL) and mixed for 15 min at a frequency of 15 Hz using a Fritsch pulverisette 23 Mini Mill. The resulting composites were used for further investigations.

## Laser flash analysis

For thermal transport measurements powder pellets were prepared. To this end, the materials were pre-compacted at 370 MPa for 3 min and subsequently isostatically pressed at 500 MPa for 60 min. Thermal diffusivities were measured in a temperature range from −100 °C to 100 °C with a step size of 25 °C under a constant nitrogen flow (100 mL min⁻¹) using a NETZSCH LFA 467. To ensure good absorption and emission properties for all samples and to protect the samples from direct contact with the atmosphere during the brief sample transfer into the LFA, carbon coatings were applied in a nitrogen-filled glovebox. In each measurement, three thermal diffusivity values (five at −100 °C) were recorded per temperature step using a MCT-detector and a ZnS sample chamber window. The mean of these values is taken as the true sample thermal diffusivity and the standard deviation between the values is taken as the measurement uncertainty. The detection time and flash intensity were automatically adjusted by the integrated software during the experiment. All measurement signals were fitted using an improved Cape-Lehmann model provided by the NETZSCH LFA Analysis 8.0.3 software to determine the respective diffusivities.

## X-ray diffraction

Powder X-ray diffraction measurements were performed to confirm phase purities of the investigated materials and the absence of sample degradation during transfer into the LFA or the LFA experiment. All samples were measured in sealed quartz capillaries on a STOE STADI P with Dectris Mythen 2X 1 K detector and Ge(111) monochromator. A $2\theta$-range from 10° to 70° was scanned with a step size of 3° in a Debye-Scherrer geometry using Cu $K\alpha1$ radiation. Pawley refinements were performed using the TOPAS-Academic V6 software package. While a modified Thompson-Cox-Hastings pseudo-Voigt function is used to describe peak shapes, a Chebyshev function with nine variables is used to describe the background. Lattice parameters, zero error, background coefficients and peak shapes were refined.

## Scanning electron microscopy and energy dispersive X-ray spectroscopy

EDX and SEM were measured to investigate sample morphology and homogeneity of LPSCl, NCM83 and their composites. The powder pellets were attached to sample holders using carbon patches. All samples were sputter coated with gold and measured on a Carl Zeiss AURIGA CrossBeam work station. An acceleration voltage of 3 kV and the In-lens detector were used for SEM measurements, whereby 20 kV and a X-Max 80 mm² detector were used as acceleration voltage and detector for the EDX measurements.

## Cell assembly

The respective materials (100 mg) were filled in airtight cell casings[55] and densified for 3 min at 370 MPa. In the case of partial electronic conductivity measurements, the steel stamps of the cell casings served as ion-blocking contacts. For partial ionic conductivity measurements, electron-blocking LPSCl (80 mg) layers were added on each side of the sample. Subsequently In-foil (chemPUR, 99.99 %, 100 µm thickness, 9 mm diameter) and Li-foil (abcr, 99.8%, 1.5 mg) were placed on each side serving as Li-reservoirs. After cell assembly a constant pressure of 50 MPa was applied, and transport measurements were conducted.

## Direct current polarization

Experiments were performed on a Metronohm Autolab PGSTAT302N in a climate chamber at a temperature of 25 °C. Constant voltages in the range from −40 mV to 50 mV were applied for 3.5 h each and the current response was measured.

## Potentiostatic electrochemical impedance spectroscopy

Measurements were conducted on a BioLogic VMP-300 at 25 °C in a climate chamber at a temperature of 25 °C. An equilibration step of 2 hours at open-circuit voltage was used before each measurement, whereas the voltage at the end of the equilibration time was used as quasi-stationary DC-voltage for the impedance measurement. In the frequency range from 7 MHz to 100 mHz, 15 data points were measured per decade of frequency with a perturbation voltage of 10 mV. Data analysis was performed using the RelaxIS3 software.

## Data availability

Data supporting the plots and findings in this work can be accessed via https://doi.org/10.17879/54928371863 and are made available by the datastore of the University of Münster[56]. Source data are provided with this paper.

## Code availability

The code for resistor network calculations can be accessed via https://doi.org/10.17879/16948580876. It is published under a MIT license and made available through the datastore of the University of Münster[57].

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

## Acknowledgements

L.K. is a member of the International Graduate School for Battery Chemistry, Characterization, Analysis, Recycling and Application (BAC-CARA), which is funded by the Ministry of Culture and Science of North Rhine-Westphalia, Germany. The authors acknowledge financial support within the cluster of competence FESTBATT funded by Bundesministerium für Bildung und Forschung (BMBF; projects 03XP0597A). Simulations were carried out on the computer cluster PALMA II at the University of Münster.

## Author contributions

L.K., T.B. and W.G.Z. conceived the presented idea, L.K. carried out the experiments with contributions of N.G., L.K analyzed the data, developed the code for the resistor network simulations and performed computations, L.K. wrote the manuscript with support from T.B. and W.G.Z. All authors discussed the results and commented on the manuscript.

## Funding

## Competing interests

The authors declare no competing interests.
