## [Peer Review File · Nature Communications]

Using resistor network models to predict the transport properties of solid-state battery composites

Corresponding Author: Professor Wolfgang Zeier

Version 0:

Reviewer comments:

Reviewer #1

(Remarks to the Author)

Reviewer's comments:

In the manuscript entitled "Using resistor network models to predict the transport properties of solid-state battery composites", the authors report on a resistor network capable of providing information on the ionic and electronic conductivity of solid electrolyte – insertion battery material composites.

I have two specific comments regarding the presented work

1. The authors used a transmission line model as reported in a 2021 paper (10.1149/1945-7111/abf8d7) to analyze the impedance results and the problem is that I do not agree with the topology of the elements in that model. Namely, a physics-based transmission line model should start with encompassing all of the processes happening in the cell (in this case, the charge transfer reaction and solid-state diffusion are missing). Then, if one is interested in only the electronic and ionic conductivities, the model is simplified by checking the capacitor elements and evaluating their contributions at very high frequencies. This way, the transmission line model can be simplified to the elements relevant for that frequency range. This was not done in the referenced paper, which only assumed no charge transfer or mass transport takes place at fully lithiated state, which I disagree with, since any perturbation will induce some changes to the cell. This TLM circuit to me looks as if the ionic charge carriers are everything but the cathode active ion. Yes, it fits the experimentally measured spectra - because it contains enough elements to do that, but I have doubt whether the values have a direct physical meaning. My suggestion would therefore be to find other relevant physics-based EIS models and cross check with those as well.

2. Just before the conclusions, the authors state that the model is not suitable to describe reaction pathway and chemo-mechanical influences, but I feel like more emphasis should be given to the limitations of the provided model. As far as I understand it, the proposed resistor network model is capable of the determination of ionic and electronic conductivities of the solid electrolyte – cathode material composites only. That is a starting point, but it rarely determines the material performance, since we get no information on the charge transfer reaction processes or the solid-state diffusion processes, which dominate the cell impedance. I therefore have reservations about the usefulness of the proposed model both for the design of new materials and testing of the existing ones.

(Remarks on code availability)

Reviewer #2

(Remarks to the Author)

The study investigated the using of resistor network (RN) model to understand the transport properties of the composite cathode having cathode active material (CAM) $\text{LiNi}_{0.83}\text{Co}_{0.11}\text{Mn}_{0.06}\text{O}_2$ and solid state electrolyte (SE) $\text{Li}_6\text{PS}_5\text{Cl}$. It is significantly important to know the composition between CAM and SE such that the utilization of the CAM is maximized. The RN model and the studies in this manuscript is insightful and will be needful for understanding transport properties in composites. However, some explanation is required to the following question

1. Authors did validation of the model by calculating the composition dependent effective conductivity for the cases of series and parallel condition as discussed in Supplementary section (S1.4). As the total effective conductivity is generally influence by the grain-boundary conductivity, the author compared RN simulation results with the analytical solution obtained by solving the equation S8 and S9 (which does not considered grain boundary condition). Authors should explain the reason

for having similar results between RN simulations (expecting to simulate by considering grain boundary condition, S3) with analytical solution (S8 and S9).

2. The plot between composition fraction with ionic conductivity as in Fig .3d (Fig.S21) obtained from RN simulation are in agreement with data obtained from EIS and TLM upto 40% CAM material. The author should explained such diversion of RN simulation data from the EIS and TLM fitting data at higher concentration where CAM material is dominant (author's assume $\sigma_{e,LPSCI} = \sigma_{ion,NCM} \approx 0 \text{ mS cm}^{-1}$). This might be related to interfacial grain boundary issues. However, author should explain it as experimentally during cell fabrication higher CAM loading material is considered.

3. The authors introduce a valuable resistor network model simulation to understand the transport phenomenon and studied the electronic and ionic conductivity properties of cathode composites. Is the model only restricting to composites of ionic and electronic to understand transport properties (direction to composites electrolyte).

(Remarks on code availability)

Reviewer #3

(Remarks to the Author)

(Remarks on code availability)

Version 1:

Reviewer comments:

Reviewer #1

(Remarks to the Author)

I have no further comments.

(Remarks on code availability)

Reviewer #2

(Remarks to the Author)

I would like to acknowledge the efforts made by the authors to address most of the comments effectively. The manuscript in its current state provides a lot of useful insights into using resistor network models to understand transport properties in composites.

However, after reviewing the revised manuscript and the responses from other reviewers, it would be helpful if the authors could comment on the scope of application of the model, which in its current state still appears limited. The manuscript does not fully explore the real-world applications of the model concerning the composition of electrodes (high loading CAM), particularly in terms of electronic and ionic conductivity, as well as thermal properties.

Further, the statement in the abstract claiming that the model "offers valuable guidance for experimentalists to streamline the tedious process of performing a multitude of experiments to understand and optimize the effective transport of composite electrodes" seems misaligned with the model's scope. In practice, solid-state batteries require high active mass loading, particularly above 70% CAM. In comparison, the model appears applicable only up to 40% CAM with effectiveness, as noted by the authors regarding the percolation limit. This can be viable only due to the assumption of $\sigma_{ionic,CAM} = 0 \text{ mS-cm}^{-1}$ during modeling, but this is not true in practice where the CAM material has both electronic as well as ionic conduction. While the model provides insights into electron and ion contributions, it may not be sufficient for optimizing experimental procedures.

I believe the manuscript will be more comprehensive in its scope if the authors could address these discrepancies as well.

(Remarks on code availability)

Reviewer #3

(Remarks to the Author)

(Remarks on code availability)

Reviewer's 1 and 3:

In the manuscript entitled "Using resistor network models to predict the transport properties of solid-state battery composites", the authors report on a resistor network capable of providing information on the ionic and electronic conductivity of solid electrolyte – insertion battery material composites. I have two specific comments regarding the presented work

Response: We thank the reviewers for their time and both very insightful comments. Before we respond to those in detail, we would like to address a general thought that may help to understand the necessity of the work of the manuscript. The more in-detail information will be addressed in the comments. Overall, solid state battery composites have two general issues that need to be addressed for solid-state batteries to be successful.

- The first is – as the reviewers point out – the challenge on chemomechanics and degradation. The processes occurring need to be measured and understood, which is not the goal of this work.
- The second general challenge is the overall transport in composites. Unless there is balanced transport, reaction fronts and inhomogeneous reactions will occur (ultimately affecting the degradation in point 1). This is especially an issue in thick electrodes and high current densities.

Whereas the first challenge is important to be addressed being it with different materials compositions and protection concepts, the second may be even more important when considering that changing compositions, particle sizes, coatings and additives will affect the partial transport. The phase space that opens is huge and needs to be screened. Our work is meant to provide a toolkit for experimentalists to narrow down the most reasonable compositions, whenever components are being changed, without the need for excessive experiments and measurements.

We hope these issues become clearer with the comments addressed and hope that it convinces the reviewers of the suitability of the manuscript.

The authors used a transmission line model as reported in a 2021 paper (10.1149/1945-7111/abf8d7) to analyze the impedance results and the problem is that I do not agree with the topology of the elements in that model. Namely, a physics-based transmission line model should start with encompassing all of the processes happening in the cell (in this case, the charge transfer reaction and solid-state diffusion are missing). Then, if one is interested in only the electronic and ionic conductivities, the model is simplified by checking the capacitor elements and evaluating their contributions at very high frequencies. This way, the transmission line model can be simplified to the elements relevant for that frequency range. This was not done in the referenced paper, which only assumed no charge transfer or mass transport takes place at fully lithiated state, which I disagree with, since any perturbation will induce some changes to the cell. This TLM circuit to me looks as if the ionic charge carriers are everything but the cathode active ion. Yes, it fits the experimentally measured spectra - because it contains enough elements to do that, but I have doubt whether the values have a direct physical meaning. My suggestion would therefore be to find other relevant physics-based EIS models and cross check with those as well.

Response: We thank the reviewers for this comment and agree that this is an important point. In the field of solid-state batteries, EIS measurements in symmetric cell setups evaluated with transmission-line models (TLM) are a common way to study effective ionic and electronic conductivities. In such symmetric setups typically either ionic or electronic charge transport is blocked. Since low frequencies correspond to long time scales and, in the limit of long timescales, only the unblocked charge carrier contributes to the overall current flow, the impedance response in the low frequency limit corresponds to the overall resistance to ionic or electronic current respectively.

Regarding the TLM usage to evaluate impedance results in this work, the objective is to evaluate the total effective ionic and electronic conductivities of the electrode composites. To extract such information, an equivalent circuit succeeding in describing the low frequency impedances and reducing to the resistance to ionic and electronic current for frequencies $\rightarrow 0$ Hz is needed. The TLM used in this work fulfils these requirements and hence succeeds in correctly extracting the desired information about effective ionic and electronic conductivities. To validate our approach, in this work each conductivity determined using the transmission line approach was further double checked by determining the effective conductivities through a DC-polarization experiment (details in supporting information S4.3).

We agree with the reviewers, that the TLM used in this is not capable of providing a complete and detailed description about the complex underlying nature of charge transport in electrode composites – being it the diffusion in the electrodes, charge transfer processes or Faradaic processes as a function of the SOC, which also means that not each of the elements can directly be converted into a physical meaning. Nevertheless, as the goal of the work is to describe the partial electronic and ionic transport, this overall approach of describing charge transport in these composites by an interconnected ionic and electronic path (TLM) is physically meaningful. Hence, choosing such a model, with elements that have been proven to well-describe impedances of this composite-class in the past ([DOI 10.1149/1945-7111/abf8d7](https://doi.org/10.1149/1945-7111/abf8d7)) and reduces to the desired resistances to ionic and electronic current for frequencies $\rightarrow 0$ Hz respectively is, from our standpoint, the most suitable equivalent circuit model for the purpose of investigating effective charge transport in the symmetric cell setups used in this work.

To phrase it differently, the TLM employed in this work should rather be viewed as a tool, that is used to correctly extract effective conductivities from the measured impedances, than a model that correctly describes the details of charge transport in composites.

To further verify the obtained results, below an alternative model (DOI 10.1021/acsaem.0c02606) that is readily used to evaluate partial transport of mixed ionic-electronic conductors (MIEC) is shown. It can fit the low frequency impedance part of a composite with $\varphi_{\text{NCM}} = 40\%$.

The results are in good agreement with the data obtained from fitting with the TLM model. A comparison of the results is shown in the table below:

	TLM	“Classic MIEC model”
$\sigma_e / \text{mS}\cdot\text{cm}^{-1}$	0.36	0.36
$\sigma_{\text{ion}} / \text{mS}\cdot\text{cm}^{-1}$	0.23	0.23

To reflect these thoughts on the employed TLM, the shortcomings and benefits of this approach the manuscript has been updated. It now reads:

“Although not providing a complete and accurate description of the complex underlying charge transport in this composite class, it well reflects many aspects of the contemporary understanding about charge transport in electrode composites and, more importantly, can be used to correctly determine the total effective conductivities from the impedance spectra.¹² Exemplary experimental results and TLM fits of a NCM83-LPSCI composite cathode with $\varphi_{\text{NCM}} = 40\%$ are shown in the

bottom of Figure 3 a and Figure 3 b, respectively. To further verify the ability of the TLM to determine the desired transport properties, in section S4.2 an alternative equivalent circuit often used for mixed ionic-electronic conductors⁴³ is used to evaluate the impedance data of the composite with $\varphi_{\text{NCM}} = 40 \%$.

We also added a small discussion in the supporting information:

“To further verify that the desired effective electronic and ionic conductivities of the composites can be correctly determined by evaluating the impedance data with the TLM, an alternative equivalent circuit, readily used to evaluate impedance results of mixed ionic-electronic conductors (MIEC) is employed in the following.¹⁶ The model can fit the low frequency impedances of a composite with $\varphi_{\text{NCM}} = 40 \%$ (Figure S12). No significant differences in the resulting ionic and electronic conductivities compared to the evaluation with the TLM model are observed, as both circuits yield $\sigma_e = 0.36 \text{ mS cm}^{-1}$ and $\sigma_{\text{ion}} = 0.23 \text{ mS cm}^{-1}$ for this composition.

Figure S1: Impedance data of a composite with $\varphi_{\text{NCM}} = 40 \%$ measured in an a) ion-blocking and b) electron-blocking setup. The equivalent circuits used for fitting are shown in the top and commonly used by the mixed ionic-electronic conductor community to evaluate effective transport properties.¹⁶

2. Just before the conclusions, the authors state that the model is not suitable to describe reaction pathway and chemo-mechanical influences, but I feel like more emphasis should be given to the limitations of the provided model. As far as I understand it, the proposed resistor network model is capable of the determination of ionic and electronic conductivities of the solid electrolyte – cathode material composites only. That is a starting point, but it rarely determines the material performance, since we get no information on the charge transfer reaction processes or the solid-state diffusion processes, which dominate the cell impedance. I therefore have reservations about the usefulness of the proposed model both for the design of new materials and testing of the existing ones.

Response: It is correct that the model does not make predictions about possible reaction pathways and chemo-mechanical influences, this is not the goal. The model and approach succeed in describing effective thermal, ionic, and electronic conductivity of the whole composite electrode. Determining and predicting effective transport in composite electrodes is the key objective of our work as it is of paramount importance for electrode design considerations.

Solid state battery composites have two general issues that need to be addressed for solid-state batteries to be successful:

1. Of course, the chemomechanics and degradations need to be monitored, understood and controlled for long-term stability. This issue we are not addressing in the current work, but we

believe that our approach can be extended in the future to include this, for instance by extending the resistor network model with additional physical processes.

2. *Charge transport needs to be balanced, otherwise thick electrodes with realistic loadings are not possible or long-term stable (point 1).* As both, electrons and ions need to access the active material during the charging and discharging reactions, fast ionic and electronic conductivity throughout the whole thickness of the composite electrode are necessary for high electrode utilizations. Furthermore, both transport quantities need to be balanced to ensure homogeneous charging and discharging reactions and to prevent the occurrence of reaction fronts during charging and discharging (DOI 10.1002/aenm.202203426). A parameter to describe the nonuniformity of the reactions taking place in a battery can be defined as

$$\delta = L \cdot |I| \cdot \beta \cdot \left(\frac{1}{\sigma_e^{\text{eff}}} + \frac{1}{\sigma_{\text{ion}}^{\text{eff}}} \right), \quad (1)$$

where L , I and β are the electrode length, the applied current and a constant related to battery chemistry (DOI abs/10.1002/aic.690210103, DOI 10.1002/aenm.202203426). For $\delta < 1$, typically homogeneous reactions throughout the whole battery thickness can be expected, while values $\delta > 1$ often result in nonuniform charging and discharging reactions with reaction fronts ultimately leading to performance limitations. To optimize a composition with respect to volume fraction, so that uniform charging and discharging reactions are possible effective conductivities hence pose a key quantity to consider.

Due to the importance of the effective transport in composites, this quantity also needs to be reconsidered when changing the composition of active material, solid electrolyte, their particle sizes or even using binders or additives. For experimentalists, this becomes a somewhat impossible challenge as the phase space is too large to test during optimization. Our proposed model can provide valuable guidance about changes that can be expected. The fact that it can be measured and modelled provides a fast toolkit for experimentalists to find optimal ratios of

composite compositions for their research. We believe this is indeed novel and quite important when optimizing solid-state battery composites.

To make the importance of the approach clearer, the text now reads:

“In these composite systems, a sufficiently high ionic (σ_{ion}) and electronic (σ_e) conductivity are necessary, given that both charge carriers need to access the active material during the charging and discharging reactions. Since a large mismatch in effective ionic and electronic conductivity leads to inhomogeneous reaction rates throughout the whole electrode thickness, balancing of the two transport quantities is of paramount importance to enable high electrode utilizations and avoid reaction fronts during battery operation.^{6,7}”

In addition, the manuscript now has an added paragraph on the limitations. We think this is an excellent point from the reviewer and fits well in the section on the benchmarking with other types of composites. The manuscript now reads:

“For more advanced research questions such as composites without well-defined microstructures or additives that cannot be simply modelled by a voxel approach (Figure S21) however, the model runs into limitations. While the current model can predict macroscopic effective total transport properties based on composite microstructures, it does not yet provide detailed microscopic insights on charge transport mechanics.”

Reviewer 2:

The study investigated the using of resistor network (RN) model to understand the transport properties of the composite cathode having cathode active material (CAM) $\text{LiNi}_{0.83}\text{Co}_{0.11}\text{Mn}_{0.06}\text{O}_2$ and solid state electrolyte (SE) $\text{Li}_6\text{PS}_5\text{Cl}$. It is significantly important to know the composition between CAM and SE such that the utilization of the CAM is maximized. The RN model and the studies in this manuscript is insightful and will be needful for understanding

transport properties in composites. However, some explanation is required to the following question

Response: We thank the reviewer for his/her time in reviewing this work and their strong support in publication.

1. Authors did validation of the model by calculating the composition dependent effective conductivity for the cases of series and parallel condition as discussed in Supplementary section (S1.4). As the total effective conductivity is generally influence by the grain-boundary conductivity, the author compared RN simulation results with the analytical solution obtained by solving the equation S8 and S9 (which does not considered grain boundary condition). Authors should explain the reason for having similar results between RN simulations (expecting to simulate by considering grain boundary condition, S3) with analytical solution (S8 and S9).

Response: As correctly pointed out, the series and parallel model do not consider grain boundary resistances. Here, for the code validation in section S1.4, the interface resistance was set to be $R_{int} = 0 \text{ m}^2 \text{ K W}^{-1}$ in the resistor network model. By doing this, Eq. S3 reduces to Eq. S2 and no grain boundary resistances are considered. We have now added this information in the Supporting Information:

“Voxel structures representing series and parallel connected materials were constructed and resistor network simulations were performed by using the experimentally assessed thermal conductivities of LPSCI and NCM83 and a thermal interfacial resistance of $0 \text{ m}^2 \text{ K W}^{-1}$ as inputs (Figure S3 b).”

2. The plot between composition fraction with ionic conductivity as in Fig .3d (Fig.S21) obtained from RN simulation are in agreement with data obtained from EIS and TLM up to 40% CAM material. The author should explained such diversion of RN simulation data from the EIS and TLM fitting data at higher concentration where CAM material is dominant (author’s assume σ_e ,LPSCI

= $\sigma_{\text{ion,NCM}} \approx 0 \text{ mS cm}^{-1}$). This might be related to interfacial grain boundary issues. However, author should explain it as experimentally during cell fabrication higher CAM loading material is considered.

Response: We thank the reviewer for this comment. With increasing volume fraction of the respective insulating phase, the system moves closer to the percolation threshold. As closer to the real percolation threshold the actual conductivities strongly depend on the microstructure (Figure 5c) ([DOI 10.1103/PhysRevE.76.041108](https://doi.org/10.1103/PhysRevE.76.041108)), we think that the predicted conductivities in that regime should be treated with caution as the microstructure of the composite is assumed by the simplified model (Section S1.1)). We think that deviations in that regime are mainly caused by the microstructure not being well enough represented. Similar, more advanced approaches ([DOI 10.1002/batt.202300167](https://doi.org/10.1002/batt.202300167)) to model transport in such composites also seem to show larger deviations to experimentally assessed values for compositions with larger volume fraction of the respective insulating phases.

We have now added an explanation to make this clearer:

“Since the percolation threshold strongly depends on the microstructure, larger deviations between measured and simulated conductivities can be expected close to the threshold as only simplified, virtual microstructures are assumed in the resistor network simulations.^{39,46} Differences in the measured and simulated effective ionic conductivities for compositions with $\varphi_{\text{NCM}} > 40 \%$ may be attributed to this effect.”

3. The authors introduce a valuable resistor network model simulation to understand the transport phenomenon and studied the electronic and ionic conductivity properties of cathode composites. Is the model only restricting to composites of ionic and electronic to understand transport properties (direction to composites electrolyte).

Response: Resistor network models to describe transport properties of porous materials and composites are also applied in other fields. The proposed model should hence not be restricted to composite electrodes only. Work from other fields of research employing similar resistor network can be read here: [DOI 10.1016/j.ijthermalsci.2012.09.008](https://doi.org/10.1016/j.ijthermalsci.2012.09.008), [DOI 10.1007/BF01020426](https://doi.org/10.1007/BF01020426), [DOI 10.1016/j.ces.2013.05.030](https://doi.org/10.1016/j.ces.2013.05.030). By also showing the dependence of porosity and polymer composites, we meant to show the broad applicability of the approach. To underline the wide application area of resistor network models, we have now added the following:

“Resistor network models are widely used in various fields such as statistical physics,⁵⁰ thermal barrier coatings⁴⁰ or in fuel cells,⁵¹ to name just a few examples.”

Reviewer 3:

(Remarks to the Author): I co-reviewed this manuscript with one of the reviewers who provided the listed reports. This is part of the Nature Communications initiative to facilitate training in peer review and to provide appropriate recognition for Early Career Researchers who co-review manuscripts.

Response: We thank the reviewer for his/her time in reviewing this work.

Reviewer's 1 and 3:

I have no further comments.

Response: We thank the reviewers for their time and their very insightful comments, which helped to improve the manuscript.

Reviewer 2:

I would like to acknowledge the efforts made by the authors to address most of the comments effectively. The manuscript in its current state provides a lot of useful insights into using resistor network models to understand transport properties in composites.

Response: We thank the reviewer for his/her time in reviewing this work.

However, after reviewing the revised manuscript and the responses from other reviewers, it would be helpful if the authors could comment on the scope of application of the model, which in its current state still appears limited. The manuscript does not fully explore the real-world applications of the model concerning the composition of electrodes (high loading CAM), particularly in terms of electronic and ionic conductivity, as well as thermal properties. Further, the statement in the abstract claiming that the model “offers valuable guidance for experimentalists to streamline the tedious process of performing a multitude of experiments to understand and optimize the effective transport of composite electrodes” seems misaligned with the model's scope. In practice, solid-state batteries require high active mass loading, particularly above 70% CAM. In comparison, the model appears applicable only up to 40% CAM with effectiveness, as noted by the authors regarding the percolation limit.

Response: We agree that solid-state batteries employ cathodes with mass loadings of ≈ 70 weight percent and our work does show this quite well. Since electronic and ionic pathways in a composite strongly depend on the volume fractions, effective transport is typically discussed in terms of volume fractions. In contrast to the usual practice in the battery community, in this work the compositions are hence specified in volume percent (φ) and not in weight percent. Using the definitions of density ($\rho_i = \frac{m_i}{V_i}$) and weight fraction ($w_i = \frac{m_i}{\sum_j m_j}$) a simple relationship between volume fraction and weight fraction can easily be derived.

$$\varphi_i = \frac{V_i}{\sum_j V_j} = \frac{m_i/\rho_i}{\sum_j m_j/\rho_j} = \frac{w_i/\rho_i}{\sum_j w_j/\rho_j}.$$

Inserting the crystallographic densities specified in the method section of this work:

“For composites, LPSCI and NCM83 were combined in the respective volume ratios whereby 1.86 g cm⁻³ and 4.78 g cm⁻³ were assumed as densities.^{54,55”}

and typical weight fractions employed in solid state-battery cathodes (e.g. 70 weight percent NCM and 30 weight percent LPSCI) weight fractions and volume fractions can be related as shown in the following example:

$$\varphi_{\text{NCM}} = \frac{0.7/4.78 \text{ g} \cdot \text{cm}^{-3}}{0.7/4.78 \text{ g} \cdot \text{cm}^{-3} + 0.3/1.86 \text{ g} \cdot \text{cm}^{-3}} = 0.48$$

In the following plot in addition to the volume fractions, the mass fractions of the investigated compositions are shown. Further, the typical weight fraction employed in solid state batteries is highlighted by a vertical line.

As our work scans the transport over different volume and weight fractions, precisely over the range of typical loadings, we believe that the resistor networks gives a valuable approximation for practical solid state battery electrodes. To further highlight that volume fractions and not mass

fractions are discussed, we have made this more explicit in the captions of the figures in the manuscript.

“Figure 3: ...Data (circles) are exemplary shown for a NCM83-LPSCI composite with a NCM volume fraction of 40 %... Resulting effective ionic- and electronic conductivities at room temperature as a function of NCM volume fraction with error bars indicating relative errors of 50 %...”

“Figure 4: ...Experimentally assessed room temperature thermal conductivities (color-coded circles) as a function of NCM volume fraction in comparison to ...”

“Figure 5: ...in comparison to simulated conductivities of the resistor network model (filled circles with dashed guide to the eye) as a function of LiMn_2O_4 volume fraction...comparison to results of the resistor network model (filled circles with dashed guide to the eye) as a function of LPSCI volume fraction...Influence of the particle size and volume fraction of an insulating second phase...”

We further added a small table to the supporting information:

S11 – Volume and weight fractions of the investigated composites

The volume ratios ($\varphi_{\text{NCM}} / \varphi_{\text{LPSCI}}$) and weight ratios ($w_{\text{NCM}} / w_{\text{LPSCI}}$) of the investigated NCM83-LPSCI composites are specified in Table S3.

Table S3: Volume fractions and weight fractions of the investigated LPSCI-NCM composites.

$\varphi_{\text{NCM}} / \varphi_{\text{LPSCI}}$	100 / 0	80 / 20	60 / 40	40 / 60	20 / 80	0 / 100
$w_{\text{NCM}} / w_{\text{LPSCI}}$	100 / 0	91 / 9	79 / 21	63 / 37	39 / 61	0 / 100

This can be viable only due to the assumption of $\sigma_{\text{ionic,CAM}} = 0 \text{ mS-cm}^{-1}$ during modeling, but this is not true in practice where the CAM material has both electronic as well as ionic conduction. While the model provides insights into electron and ion contributions, it may not be sufficient for optimizing experimental procedures.

Response: We disagree with the statement that the model is only viable when assuming one of the conductivities to be zero. To highlight this, we added a further case study to our model validation section: Yoshio Yuge simulated effective conductivities of binary composites on a simple cubic lattice and derived a modified effective medium theory that is designed explicitly for the case when both phases exhibit different conductivities

(<https://doi.org/10.1007/BF01020426>). His research question can also be studied using the resistor networks presented in this work when building resistor networks from voxel structures where no clustering has been applied. In his examples, Yuge calculated effective conductivities for cases where both phases exhibit different. We are able to reproduce these results, showing that the model can also yield valuable information when both phases add to the total conductivity. We hope this additional case study – in combination with the ones shown for a variety of different cases – convince the reviewer of the broad validity to understand electronic, ionic and thermal transport in electrode composites.

The supporting information now reads:

Figure S1: a) A representative progression of the residual value with the number of iterations. b) Effective conductivities expected from the analytical solutions for the series and parallel model (lines), compared to effective conductivities simulated using resistor networks representing in parallel and in series connected materials respectively (circles). c) Modified effective medium theory reported by Yuge (lines),⁵ compared to effective conductivities calculated using our Resistor network model (circles) with different conductivities of solid electrolyte and cathode active material (CAM). No clustering has been applied when constructing the 30 x 30 x 30 voxel structures to match the conditions described by Yuge.⁵

... To further validate the code and to highlight its ability to reliably predict effective composite conductivities when conductivities of e.g. cathode active material and solid electrolyte are close to each other, numerical results by Yuge have been reproduced using the resistor network presented in this work and are shown in Figure S3 c.⁵ Voxel structures were constructed using $L_{\text{vox}} = 30$ and $S_{\text{clust}} = 1$, while the conductivities have

been varied according to the conditions reported by Yuge.⁵ The author also derived a modified effective medium theory (EMT) for conductivities of binary composites on a simple cubic lattice.⁵

$$\sigma_{\text{eff}} = \frac{1}{4} \cdot \left[(3\varphi_1 - 1)\sigma_{\text{m1}} + (3\varphi_2 - 1)\sigma_{\text{m2}} + \sqrt{[(3\varphi_1 - 1)\sigma_{\text{m1}} + (3\varphi_2 - 1)\sigma_{\text{m2}}]^2 + 8\sigma_{\text{m1}}\sigma_{\text{m2}}} \right] \quad (\text{S10})$$

with $\sigma_{\text{m1}} = \varphi_1\sigma_1 + \varphi_2 \cdot \frac{2\sigma_1\sigma_2}{\sigma_1 + \sigma_2}$ and $\sigma_{\text{m2}} = \varphi_2\sigma_2 + \varphi_1 \cdot \frac{2\sigma_1\sigma_2}{\sigma_1 + \sigma_2}$. The numerical results show good agreement with Eq. S10.⁵